# Pharmacological Mechanism and Drug Research Prospects of Ginsenoside Rb1 as an Antidepressant

**DOI:** 10.3390/antiox14020238

**Published:** 2025-02-19

**Authors:** Shuhui Zhuang, Fuqiang Shi, Nazzareno Cannella, Massimo Ubaldi, Roberto Ciccocioppo, Hongwu Li, Di Qin

**Affiliations:** 1School of Chemical Engineering, Changchun University of Technology, Changchun 130012, China; 2202206099@stu.ccut.edu.cn (S.Z.); shifq@ccut.edu.cn (F.S.); 2Pharmacology Unit, School of Pharmacy, University of Camerino, Via Madonna delle Carceri 9, 62032 Camerino, Italy; nazzareno.cannella@unicam.it (N.C.); massimo.ubaldi@unicam.it (M.U.); roberto.ciccocioppo@unicam.it (R.C.); 3Department of Geriatrics and General Practice, The Third Bethune Hospital of Jilin University, Changchun 130021, China; q148089882@jlu.edu.cn

**Keywords:** depression, major depressive disorder, ginsenoside Rb1, antidepressant mechanisms, natural ingredients, inflammation

## Abstract

This review explores the antidepressant effects of ginsenoside Rb1, a natural compound in traditional Chinese medicine, and its potential for treating major depressive disorder (MDD). The aetiology of depression was reviewed up to 2024, focusing on the pathways and mechanisms through which ginsenoside Rb1 may exert its effects. Notably, ginsenoside Rb1 regulates oxidative stress and inflammatory processes while enhancing neural plasticity by downregulating miR-134 expression and alleviating depressive symptoms. Unlike traditional antidepressants that act on a single target, ginsenoside Rb1 interacts with multiple pathways, reflecting its potential for broader therapeutic application. To compensate for the current deficiency in animal experiments, clinical data, and research on the side effects of ginsenoside Rb1 in the treatment of depression, we reviewed some clinical data on the use of this component in the treatment of other diseases to explore its relevance to depression. Ginsenoside Rb1 is expected to serve as a novel antidepressant or as a complementary component in combination with other antidepressant compounds. However, further clinical trials and molecular studies are necessary to confirm its efficacy and potential side effects.

## 1. Introduction

Major depressive disorder (MDD) is a pervasive mood dysfunction characterised by persistent spontaneous low mood [1,2], affecting >350 million people globally in 2023 (Personality & Social Psychology Bulletin, 2023), with a growth rate of approximately 18% over the past decade according to the World Health Organisation. According to a Chinese mental health survey, >95 million people in China have depression. The percentage of the population that has experienced depression at some point in their lives is 6.8%, whereas that of people who have experienced depression during the past 12 months is 3.6%. MDD is characterised by impaired cognition, mood regulation, memory, motor function, motivation, and neurotrophic symptoms and often leads to sleep disturbances, loss of appetite, and severe disability. It can lead to secondary disabilities, as individuals with depression are more likely to develop chronic illnesses and less likely to adhere to treatment. The combination of primary and secondary disabilities associated with depression and chronic illnesses makes MDD one of the most expensive healthcare burdens worldwide [3,4]. Despite its prevalence and considerable impact, the mechanisms of its pathogenesis and recovery are relatively unclear [5,6] owing to the involvement of various causative factors (biological, psychological, and social) and feedback loops compared with other common chronic and potentially fatal multifactorial disorders [7,8].

Amitriptyline, clomipramine, and doxepin, among other tricyclic antidepressants, influence the serotonin levels (5-hydroxytryptamine; 5-HT) in the brain and restore balance by increasing the supply of monoamine neurotransmitters. Amitriptyline is a first-generation antidepressant [9]. In the 1970s, researchers thought that effective antidepressants acted on monoamine neurotransmitters based on the ‘monoamine hypothesis’ [10] and discovered the effects of 5-HT and dopamine (DA) on mood. This discovery led to the introduction of fluoxetine (Prozac) [11], which acts as a selective 5-HT reuptake inhibitor and exerts its antidepressant effect by blocking 5-HT reuptake and increasing the 5-HT reserves of the brain. Fluvoxamine, sertraline, paroxetine, citalopram, and other selective 5-HT reuptake inhibitors have also been used. The other class includes 5-HT norepinephrine reuptake inhibitors, viz. venlafaxine, duloxetine, and milnacipran. These have fewer side effects than first-generation antidepressants [12,13,14]. However, all the aforementioned agents are associated with side effects, such as sudden nausea and headache, persistent sexual dysfunction, weight gain, and decreased rapid eye movement sleep [15,16]. Contrarily, Chinese medicines have multiple chemical components, targets, and pharmacological effects that work synergistically while avoiding the side effects of a single-action drug. Based on research progress over the past two decades on the antidepressant effects of Chinese medicines, these appear to exert their effects through various mechanisms, including increasing synaptic monoamine concentrations, attenuating hypothalamic–pituitary–adrenal (HPA) axis dysfunction, mitigating neural plasticity damage, and counteracting immune and inflammatory dysregulation. Active antidepressant substances can be classified into general categories of saponins, flavonoids, alkaloids, and polysaccharides. Ginsenosides, namely ginsenoside Rb1, ginsenoside Rg1, and ginsenoside Rg3, are the most widely used saponins, and all of them have been shown to exert different degrees of antidepressant effects in recent studies [17].

Compared with ginsenosides Rg1 and Rg3, ginsenoside Rb1 acts and regulates different regions when exerting antidepressant effects through antioxidant and anti-inflammatory pathways. Moreover, the discovery of the antidepressant effects of its metabolite F2 indicates that there is still room for in-depth research on ginsenoside Rb1 in the field of antidepressant treatment [18]. This includes activating the nuclear factor erythroid-2-related factor 2/heme oxygenase 1(Nrf2/HO-1) and nuclear factor erythroid-2-related factor 2/antioxidant response element (Nrf2/ARE) signalling pathways, reducing the contents of reactive oxygen species (ROS) and malondialdehyde (MDA), and decreasing inflammatory factor expression. This article reviews the potential mechanisms through which the antidepressant ginsenoside Rb1 exerts its effects in the treatment of depression. Additionally, we analysed data from previous animal studies and clinical trials involving this compound in the treatment of other diseases and explored its pharmacokinetics, bioavailability, and pharmacodynamics in the context of depression. Furthermore, research on gut microbiota modulation and the combined administration of ginsenoside Rb1 with other components are expected to become key directions for improving its bioavailability.

## 2. Ginsenosides

The primary pharmacologically active ingredients in ginseng are a family of naturally occurring steroidal glycosides and triterpenoidal saponins called ginsenosides. The primary pharmacologically active ingredients in *Panax ginseng* C.A. Mey. (Chinese ginseng) are Rb1, Rb2, Rb3, Rd, and Rg1. These components can have various pharmacological effects, such as blood pressure regulation, cardiovascular protection, anti-oxidant, anti-cancer, anti-inflammatory, and anti-aging effects [19,20,21]. Ginsenosides have been found to potentially ameliorate depression through various mechanisms as novel antidepressants [22]. For example, the anxiety-like effects of ginsenoside Rg3 are involved in HPA axis dysfunction, neurosteroid biosynthesis, and serotonergic system normalisation [23]. Reportedly, Rg1 may reduce connexin 43 (Cx43) phosphorylation and inhibit hemichannel opening to ameliorate CORT-induced astrocytic glutamatergic dysfunction and ameliorate gap junction dysfunction. Conversely, studies on ginsenoside Rb1 have shown that monoaminergic (1-hydroxytryptaminergic, noradrenergic, and dopaminergic) and amino acid (glutamatergic and GABAergic) receptors may be involved in the antidepressant-like effects of Rb1 [24]. Therefore, ginsenoside Rb1 has broad research potential.

## 3. Ginsenoside Rb1

Ginsenoside Rb1 is an active substance extracted from the dried root of ginseng (family Araliaceae), belonging to the tetracyclic triterpenoid saponins (molecular formula: C_54_H_92_O_23_), which are abundant in the stem, root, and buds of *Panax ginseng* and have neuroinhibitory and stabilising effects in several mammalian models. The structure of ginsenoside Rb1 is shown in Figure 1. Ginsenosides have a similar hydrophobic tetracycline structure; however, they differ in the number of sugar groups. This variability in sugar composition may be related to the specific actions of each ginsenoside. From a chemical structure perspective, its structural features may be relevant in an antidepressant context. Its parent nuclear structure is of the dammarane type, a structure that helps ginsenoside Rb1 regulate signalling pathways in the central nervous system and may exert antidepressant effects by affecting neurotransmitter levels, such as regulating the release and reuptake processes of neurotransmitters, including 5-HT and DA. Additionally, Rb1 has pharmacological effects, such as maintaining blood circulation, improving myocardial ischaemia, and exhibiting anti-arrhythmic, neuroprotective, anti-aging, anti-oxidant, anti-obesity, and anti-tumour properties [19,21,25,26].

### 3.1. Possible Antidepressant Pathways of Ginsenoside Rb1

#### 3.1.1. Decreased Amount of NLRP3 (Inflammatory Vesicles) Translation and Activated SIRT1, HO-1, and Nrf2 Translation

Rb1 inhibits the activation of the NLRP3 inflammatory vesicle, which in turn decreases the amplification and propagation of inflammatory signals and alleviates inflammatory responses [27]. It also decreases the expression of NLRP3 inflammatory vesicle-associated proteins, such as NLRP3, apoptosis-associated speckled protein, cleaved caspase-1, and IL-1β in the hippocampus of CSDS mice, where it controls the expression of haem oxygenase-1 (HO-1) and nuclear factor E2-related factor 2 (Nrf2), reverses the downregulation of Nrf2 and HO-1 expression caused by CSDS, activates the Nrf2/HO-1 signalling pathway, and improves cellular antioxidant defences. These actions help alleviate the neuronal damage brought on by oxidative stress, which in turn improves depression-like behaviour. Furthermore, following Rb1 therapy, sirtuin 1 (SIRT1) expression in the dentate gyrus (DG) area of the mouse hippocampal region increases, as does the protein quantity and its relative fluorescence intensity. This implies that Rb1 may either directly interact with the SIRT1 protein or have an impact on the transcription and translation of the gene, which would raise SIRT1 levels. SIRT1 is implicated in aberrant emotional responses in response to stressful events, including depression, and is linked to oxidative stress in neuroinflammation [28]. SIRT1 plays a crucial function in controlling oxidative stress and inflammatory processes by regulating many transcription factors, such as NF-κB and TNF-α [29].

According to a different study [30], Rb1 can attach to particular molecules or receptors, which sets off a chain of intracellular reactions. This causes Nrf2 to move from the cytoplasm to the nucleus, where it attaches to antioxidant response elements (AREs), starting the transcription of antioxidant genes such as HO-1, ultimately activating the endothelial nitric oxide synthase (eNOS)/Nrf2/HO-1 signalling pathway.

#### 3.1.2. Reduced Reactive Oxygen Species (ROS) Content

HO-1 and Nrf2 are considered key targets for antioxidants in the body. By triggering the associated antioxidant enzymes, activation of this pathway might improve the capacity of cells to scavenge ROS, preserve redox equilibrium, and reduce oxidative damage. By increasing glutathione (GSH) levels and superoxide dismutase (SOD) activity and decreasing ROS and thiobarbituric acid reactive substances, Rb1 drastically reduces the redox imbalance and mitochondrial dysfunction caused by rotenone-induced oxidative stress in SH-SY 5 Y cells, according to in vitro studies [30]. Additionally, Rb1 increases the expression of Bcl-2 and decreases apoptosis as well as the levels of caspase-3 and Bax [31].

Significant improvements in hindlimb functional scores, a decrease in MDA content, an increase in the SOD, catalase (CAT), and GSH levels, and the upregulation of eNOS/Nrf2 and NAD (P)H quinone dehydrogenase 1 protein expression are among the major protective effects that Rb1 exerts against oxidative stress injury in rat spinal cords in vivo by activating the eNOS/Nrf2/HO-1 signalling pathway [32]. By activating the Nrf2/ARE signalling pathway, Rb1 protects neurons against Mg^2+^ free-radical-induced neuronal injury and pentylenetetrazol-induced oxidative stress [33].

Rb1 enhances the expression of HO-1 and Nrf2 in rat hippocampal tissue both in vitro and in vivo. Its main mechanism of action entails activating the Nrf2/HO-1 and Nrf2/ARE signalling pathways, increasing SOD and CAT activities, upregulating the levels of eNOS, GSH, and HO-1, decreasing the levels of ROS and MDA to improve the intracellular redox state, and reducing the expression of pro-apoptotic genes and the release of inflammatory factors in rat neuronal cells and the damage caused by oxidative stress on the same cells. 

Varying degrees of oxidative damage and antioxidant enzyme degradation are present in patients with depression, and the use of antioxidant compounds, such as N-acetylcysteine, may reduce depressive symptoms in these individuals [34]. 

Therefore, we hypothesise that ginsenoside Rb1 can restore the balance between ROS and the antioxidant defence system by reducing ROS content to achieve an antidepressant effect.

#### 3.1.3. Reduced Expression of Inflammatory Factors

In an alcohol-treated zebrafish model, ginsenoside Rb1 therapy resulted in lower levels of TNF-α and nuclear factor-κB (NF-κB) protein expression [35]. Ginsenoside Rb1 therapy led to lower mRNA expression levels of TNF-α and IL-1β in the ethanol-exposed group, according to quantitative real-time PCR.

By lowering alcohol-induced elevated NF-κB levels and the expression of its downstream inflammatory factors TNF-α and IL-1β and simultaneously inhibiting neutrophil infiltration, ginsenoside Rb1 plays a notable role in the treatment of alcoholic liver disease and prevents additional inflammatory damage to the liver caused by alcohol. Additionally, this function is essential in the treatment of depression. Further, ginsenoside Rb1 inhibits the over-activation of microglia, decreases the expression of ionic calcium-binding articulation molecule 1 (Iba1) in the hippocampal region of CSDS-exposed mice, and decreases the number of activated microglia [27]. This has been shown morphologically as a reduction in the enlargement of the cell body and coarsening and densification of protrusions.

#### 3.1.4. Regulation of miR-134 Expression

Similarly, miR-134 levels in the blood of patients with depression fluctuate and are correlated with the response to antidepressant treatment. For instance, plasma miR-134 levels increased following eight weeks of individualised antidepressant treatment after deterioration in untreated patients with depression, indicating that miR-134 may be a viable indicator of antidepressant treatment efficacy [36].

Rb1 was found to suppress the expression of miR-134, and mmu-miR-134 was shown to directly bind to BDNF 3'UTR in the hippocampus of mice exposed to chronic unpredictable mild stress (CUMS). Moreover, miR-134 overexpression prevented the Rb1 neuroprotective impact, while Rb1 enhanced the amount of NeuN-immunoreactive neurons in the hippocampal CA1, CA3, and DG regions and reduced neuronal damage in mice exposed to CUMS. In the hippocampus of mice exposed to CUMS, Rb1 markedly increased BDNF and its downstream proteins (such as TrkB, AKT, ERK1/2, GSK-3b, β-catenin, and CREB), indicating that the BDNF signalling pathway was activated. The inhibition of Rb1 activation of the BDNF signalling cascade pathway viamiR-134 overexpression implies that the antidepressant-like effects of Rb1 depend on the miR-134-mediated BDNF signalling pathway [37].

### 3.2. Ginsenoside Rb1 Exerts Its Antidepressant Effects Through Specific Loci of Action

#### 3.2.1. Ginsenoside Rb1 Stabilises the Binding of Glutathione Reductase (GR)

According to molecular docking studies, the flavin adenine dinucleotide (FAD) binding domain of ginsenoside Rb1 is where Rb1 binds to GR. Van der Waals force interactions with arginine 347, arginine 37, proline 368, threonine 339, valine 370, and lysine 66 in GR comprise the primary mechanism through which Rb1 binds to GR. According to molecular docking studies, the binding energy between GR and Rb1 is 26,913.264 J/mol, suggesting that the two have a specific binding capacity. The stability of the binding has been further confirmed using molecular dynamics simulations. When Rb1 is first coupled to GR, it is situated deep within the ligand binding site; however, after 100 ns of simulation, it migrates to the edge of the binding region [38]. 

The heavy atom root mean square deviation (RMSD) of the GR-Rb1 complex varies over the course of the simulation, with a higher trajectory than that of the unbound GR. This suggests that the structure of the complex is relatively stable after formation and that Rb1 can form a favourable conformation by binding to GR in a stable manner.

These results imply that ginsenoside Rb1 can efficiently bind to and stabilise GR, giving it a structural foundation to function as a possible GR agonist, achieving its pharmacological benefit of preventing oxidative stress and apoptosis. Additionally, glutathione disulfide (GSSG) can be converted by GR into GSH, which lowers intracellular ROS generation and may help alleviate depression.

#### 3.2.2. Ginsenoside Rb1 Activates PPARγ

Under normal circumstances, microglia exhibit two polarised phenotypes: M1 (pro-inflammatory) and M2 (anti-inflammatory). In animal models, chronic mild stress (CMS) induces a shift of microglia towards the M1-like phenotype, characterised by enlarged cell bodies, shortened and thickened processes, increased expression of pro-inflammatory cytokines, and decreased expression of anti-inflammatory cytokines. In a study using a CMS-induced depression model, GRb1 modulated microglial phenotypes by activating PPARγ, promoting microglia transition towards the M2-like phenotype (Figure 2). This shift resulted in reduced areas of Iba1+ cells in the hippocampus, increased numbers and total lengths of their processes, along with a decreased release of pro-inflammatory cytokines (such as TNF-α and IL-1β) and an increased release of anti-inflammatory cytokines (such as TGF-β and Arg-1). In vitro experiments demonstrated that GRb1 inhibited lipopolysaccharide (LPS)-induced microglia activation, reducing the areas of Iba1+ cells and the lengths of their processes. This effect was blocked by the PPARγ antagonist GW9662, effectively proving the mechanism of action of ginsenoside Rb1 [39].

#### 3.2.3. Ginsenoside Rb1 Alleviates the Decrease in γ-Aminobutyric Acid (GABA) Levels

According to the GABAergic theory of major depressive illness, changes in GABAergic transmission indicate a change in how the brain functions. Patients with depression have low levels of GABA in their plasma and cerebrospinal fluid. Significantly lower numbers of GABA receptors and lower levels of GABA release have been observed in the frontal cortex and hippocampus, respectively, of sad rodents [40].

Rb1 and *Lactobacillus helveticus* preconditioning attenuate ischaemia/reperfusion-induced reductions in protein and mRNA levels of GABA_A_ (α2, β2, and γ2) and GABA_B_ (1b and 2) receptor subunits. However, the effect of Rb1 on GABA_B_ receptor subunits is not as significant as its effect on GABA_A_ receptor subunits (especially α2 and γ2). This effect may be useful in the treatment of depression [41].

In another study, the effect of ginsenoside Rb1 was evaluated in terms of the content of monoamine neurotransmitters in the rat brain after treating rats with the CUMS model and was compared with the traditional antidepressant fluoxetine. We can intuitively observe the influence of ginsenoside Rb1 on the content of 5-HT, norepinephrine, dopamine, and GABA in the hippocampus and prefrontal cortex of rats compared with those of the traditional antidepressant fluoxetine. The experiment shows that the contents of 5-HT, norepinephrine, dopamine, and GABA are positively correlated with the ginsenoside Rb1 content. Moreover, when the dosage of fluoxetine is 10 mg/kg, the measured data in rats after fluoxetine treatment are consistently higher than those observed after treatment with ginsenoside Rb1 at the same dosage [42]. In other studies, after rats were treated with 16 mg/kg ginsenoside Rb1, the content of serotonin, norepinephrine, and dopamine in their serum was basically the same as it was after treatment with 10 mg/kg ginsenoside Rb1 (Table 1) [43]. However, based on this criterion, the effect of ginsenoside Rb1 is weaker than that of traditional antidepressants. Nevertheless, only a limited amount of data is currently available, and additional experiments are required to prove the overall value of this component.

### 3.3. Potential Antidepressant Effects Exhibited by Ginsenoside Rb1 in the Treatment of Other Diseases

#### 3.3.1. Ginsenoside Rb1 Attenuates Alcohol-Induced Liver Injury

Ginsenoside Rb1 reduced alcohol intake-induced ROS formation and mitigated oxidative stress injury in investigations of its protective action against oxidative stress in zebrafish litters and L-O2 cells following alcohol exposure [35]. Experiments with L-O2 cells in vitro yielded the same outcomes. The increase in glutathione levels following ginsenoside Rb1 injection supports the fact that glutathione can lower intracellular ROS by reacting with certain ROS. Furthermore, as previously reported, oxidative stress is an important factor in the pathophysiology of depression, adversely influencing hormone levels (such as cortisol) and neurotransmitter production.

#### 3.3.2. Ginsenoside Rb1 for the Treatment of Obesity

While nearly half of obese adults experience long-term activation of the HPA axis (hypercortisolised obesity), and children with high levels of cortisol for an extended period are at an increased risk of becoming obese, excessive activation of the HPA axis results in the sustained release of cortisol, which over time can cause neuronal damage in the limbic system and is linked to the onset of depression. By enhancing appetite, encouraging adipogenesis, and reducing energy expenditure, high cortisol levels cause obesity. Concurrently, inflammation linked to obesity disrupts cortisol receptor function, influences the activities of their metabolic enzymes, and impacts the HPA axis, creating a vicious cycle.

Chronic low-grade inflammation is linked to obesity. Pro-inflammatory cytokines released by adipocytes can cause inflammation in the brain through varied pathways, impacting neuronal damage, depression-related neurotransmitter transmission, and other pathological processes. For example, cytokine-induced activation of indoleamine 2,3-dioxygenase can result in abnormal tryptophan metabolism, neurotoxic product production, increased excitotoxicity, and decreased neurotrophic factor synthesis [44].

Both obesity and depression are associated with upregulation of inflammatory vesicles and their aberrant regulation is linked to inflammatory activation. For example, the adipocytes of individuals with obesity and peripheral-blood mononuclear cells of patients with depression express more NLRP3 inflammatory vesicles and caspase-1, and inhibiting caspase-1 alleviates the symptoms of both conditions. Additionally, NLRP3 inflammatory vesicles may influence HPA axis function by cleaving the GR HPA axis operation. Inflammatory markers, such as C-reactive protein (CRP), are elevated in subgroups of patients with depression with enhanced neurovegetative symptoms (e.g., weight gain and increased hunger) and are linked to genetic risk factors, according to clinical investigations.

Based on the studies cited thus far, we anticipate that studies on the role of ginsenoside Rb1 in managing obesity will yield pertinent data for the treatment of depression, which merits consideration.

#### 3.3.3. Clinical Studies on Ginsenoside Rb1 in the Treatment of Chronic Kidney Disease

Unfortunately, we were unable to find clinical trial data directly assessing the efficacy of ginsenoside Rb1 in the treatment of depression in previous research reports. This may be attributed to the complex pathogenesis of depression and the challenges associated with studying ginsenoside Rb1, given its ability to act on multiple targets. However, data from clinical trials investigating the effects of ginsenoside Rb1 in other diseases provide some promising insights.

Clinical trials on the use of ginsenoside Rb1 in the treatment of chronic kidney disease have demonstrated that administering 500 mg of this compound daily effectively alleviates oxidative stress and reduces systemic inflammation [45]. Most participants receiving ginsenoside Rb1 exhibited signs of reduced oxidative stress. After six months of continuous oral administration, ROS levels decreased, glutathione peroxidase activity significantly increased, and DNA damage was reduced. Moreover, when participants were reassessed six months after discontinuing the treatment, these effects remained significant. Compared with the placebo group, the ginsenoside Rb1-treated group exhibited lower TNF-α and IL-6 levels at the end of the treatment. These findings are consistent with previous animal studies on ginsenoside Rb1 for the treatment of depressive symptoms. Furthermore, the observed modulation of oxidative stress and inflammatory markers supports the potential use of ginsenoside Rb1 as an antidepressant for the treatment of depressive disorders.

### 3.4. Comparison of Ginsenoside Rb1 with Traditional Antidepressant Drugs and Other Antidepressant Ingredients

Compared with the antidepressant drugs that act on a single link (Table 2), ginsenoside Rb1 can act on multiple targets, such as regulating nerve growth factors and acting as an antioxidant and anti-inflammatory substance, thereby enhancing its comprehensive role in treatment. Traditional antidepressants are associated with several side effects, such as dry mouth, constipation, and sexual dysfunction. However, ginsenoside Rb1, being a natural product, may have a relatively milder impact and fewer side effects, making it a safer option. Long-term use of certain traditional antidepressants may burden organs, such as the heart and liver. Ginseng and its components are typically considered to have a high safety profile when used in moderation. Furthermore, as ginsenoside RB1 improves cognitive function, this property may be an additional positive effect for patients with depression.

### 3.5. Biological Properties of Ginsenoside Rb1

The disadvantages of ginsenoside Rb1, such as low water solubility and poor bioavailability, limit its wide application. Experimental studies on rats have clearly indicated that the oral bioavailability of ginsenoside Rb1 is generally <5% [80,81]. These results intuitively reflect the poor absorption of Rb1 after oral administration, mainly attributed to the relatively complex chemical structure of Rb1, which contains multiple sugar moieties. Such a structure poses numerous difficulties for its absorption in the gastrointestinal tract. One possible strategy to overcome this issue is to use materials based on nanoparticles (NPs) as delivery carriers. Additionally, the influence of the gut microbiota is an undeniable factor in modulating the absorption rate of ginsenoside Rb1.

#### 3.5.1. Improving Bioavailability Through Nanoparticle (NP) Delivery Systems

Researchers prepared mannosylated bovine serum albumin drug-loaded NPs (Man-BSA@Rb1 NPs) [82]. By encapsulating Rb1 within these nanoparticles, their solubility is effectively improved. Experimental results show that the encapsulation efficiency of these nanoparticles is as high as 96.7 ± 6.5%. Moreover, in cell uptake experiments, mannosylation enables a significant increase in the accumulation of NPs within cells, indicating that Rb1 can enter cells and exert its effects. In in vivo experiments, Man-BSA@Rb1 NP accumulation in the liver was significantly higher than that of unmodified NPs, which further proves that modified NPs can improve the distribution and utilisation efficiency of Rb1 in the body, thereby enhancing its bioavailability.

#### 3.5.2. The Influence of Gut Microbiota on Bioavailability

The gut microbiota is an important participant in the in vivo metabolism of ginsenoside Rb1. Following oral administration of Rb1 to rats, the gut microbiota metabolised it into deglycosylated metabolites such as ginsenoside Rd, ginsenoside F2 (F2), ginsenoside Rg3, compound K (CK), and 20(S)-protopanaxadiol (PPD); of these, Rd was the major metabolite [80,83]. Bacteria, such as *Bacteroides* and *Lactobacillus*, in the gut can produce a variety of enzymes, such as β-glucosidase, which can promote the deglycosylation reaction of Rb1. The relative abundance of *Bacteroides cellulosilyticus* is increased, which is significantly correlated with the pharmacokinetic parameters of Rb1. Further, the gut microbiota may participate in the metabolic process of Rb1 through its rich carbohydrate enzyme activity, change the structure of Rb1, and thus affect its absorption and distribution in and elimination from the body, ultimately affecting its bioavailability. 

Meanwhile, the gut microbiota can alter the gut environment and thus affect Rb1 absorption. The gut microbiota participates in multiple metabolic pathways. After high-dose Rb1 intervention, galactose metabolism and other glycan degradation pathways are significantly enriched in the rat gut. These pathways involve a variety of glycosidases, such as EC3.2.1.20 and EC3.2.1.22. Changes in the activities of these enzymes will affect the metabolism of carbohydrates in the intestine [83,84,85]. 

In addition, the presence of the gut microbiota can maintain the integrity of the gut barrier, affect the function and transport capacity of gut epithelial cells, and indirectly affect the absorption of Rb1 [86,87]. This perspective may provide new research directions for improving the bioavailability of ginsenoside Rb1.

### 3.6. Combination with Other Herbal Ingredients to Improve Efficacy

A Chinese compound prescription (CCP) refers to the modern pharmacology field where a Chinese herbal formula consists of multiple herbs. Their complex chemical compositions enable them to exert a more comprehensive effect on multiple pathologies or target diseases for intermittent efficacy, forming the material basis for preventing, diagnosing, and treating diseases. Additionally, the obvious advantages of drug combinations have been proven in several studies. One notable example is the WHO-recommended artemisinin-based combination therapy for the treatment of malaria and the treatment and prevention of HIV using ‘cocktail therapy’ [17].

CCPs exhibit antidepressant effects, typically derived from medicinal substances, such as Chaihu (*Bupleurum chinense*), peony, licorice (*Glycyrrhiza uralensis*), and ginger (*Zingiber officinale* Roscoe). In CCPs, these key medicinal ingredients are commonly referred to as the ‘sovereign herb’ (the herb that plays the most important role in the treatment of the primary symptoms). Therefore, we hypothesise that components such as chaihu saponin (from chaihu), paeoniflorin, glycyrrhizin, isoglycyrrhizin (present in licorice), ginsenosides (mainly present in ginseng), and curcumin (from ginger) might be key antidepressant components in CCPs. The discovery of antidepressant drugs is currently at a crucial juncture [22,88,89]. CCPs, offering the benefits of early onset, low addiction risk, high remission rates, fewer adverse effects, and long-lasting antidepressant effects, may inspire the future development of antidepressant drugs. However, the lack of scientific and technological pathways limits their development. Therefore, resolving uncontrolled quality issues and investigating CCPs at the molecular and systemic levels are cardinal to determining whether modernising them will succeed in satisfying the expanding needs of the pharmaceutical industry.

## 4. Conclusions and Future Directions

Ginsenoside Rb1 exerts a positive effect on the treatment of depression by interacting with various signalling pathways and transcription factors. The ability to stably bind glutathione reductase supports its ability to reduce ROS in cells and thus reduce the risk of oxidative stress. This makes it possible for ginsenoside Rb1 to treat depression through multi-target modulation. However, its low bioavailability remains an unresolved issue. Therefore, it is important to explore more effective delivery methods to improve bioavailability. Notable examples include NP delivery technology and the modulation of the gut microbiota. However, current research mainly focuses on proving the antidepressant potential of ginsenoside Rb1, with limited studies exploring its novel therapeutic mechanisms, structural characteristics, and the specific pathways through which it exerts its effects. Additionally, the precise activity of this compound remains insufficiently elucidated [18,90]. Most existing studies have focused on traditional therapeutic processes, including anti-inflammatory effects, which failed to effectively reflect the advantages of the overall multi-target regulation of ginsenoside Rb1. This aspect should be better elucidated. In addition, studies on the exact link between the chemical structure of ginsenoside Rb1 and its antidepressant effects are lacking, which could pose a major obstacle to the development of ginsenoside Rb1 antidepressant drugs. These limitations should be addressed. Owing to this, investigating the antidepressant effects of this constituent using cutting-edge modern methods by establishing consistent diagnostic and efficacy evaluation criteria is postulated to substantially contribute to the expanding needs of the pharmaceutical industry and provide a scientific basis for new approaches to treat depression.

In the future, research on ginsenoside Rb1 can be directed towards the combined treatment of depressive symptoms caused by other diseases, such as the depressive behaviour brought on by diabetes. Moreover, ginsenoside Rb1 can be used in combination with other natural ingredients that complement its mechanism of action for treating depression. However, the safety of such drug combinations requires further research. In addition, research data on the side effects produced by Rb1 are lacking in existing studies. In future research, the impact of ginsenoside Rb1 on blood and gut microbiota can be simultaneously detected to further improve the research on this component. Overall, ginsenoside Rb1 holds great promise in clinical applications as a potential antidepressant drug.

## Figures and Tables

**Figure 1 antioxidants-14-00238-f001:**
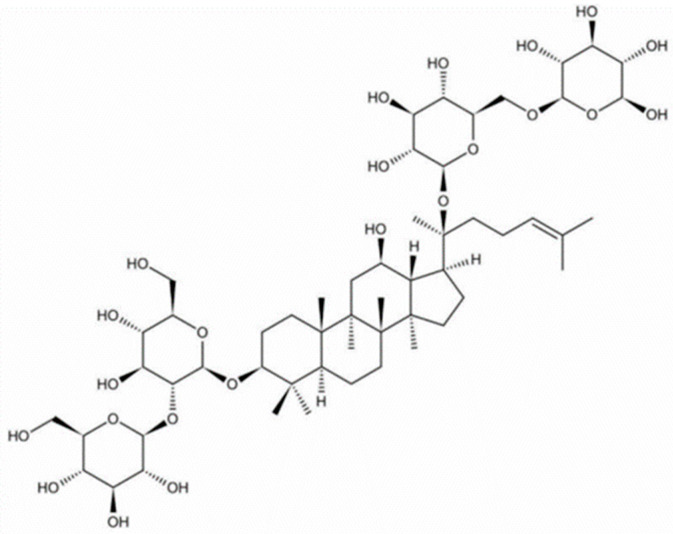
Structure of ginsenoside Rb1.

**Figure 2 antioxidants-14-00238-f002:**
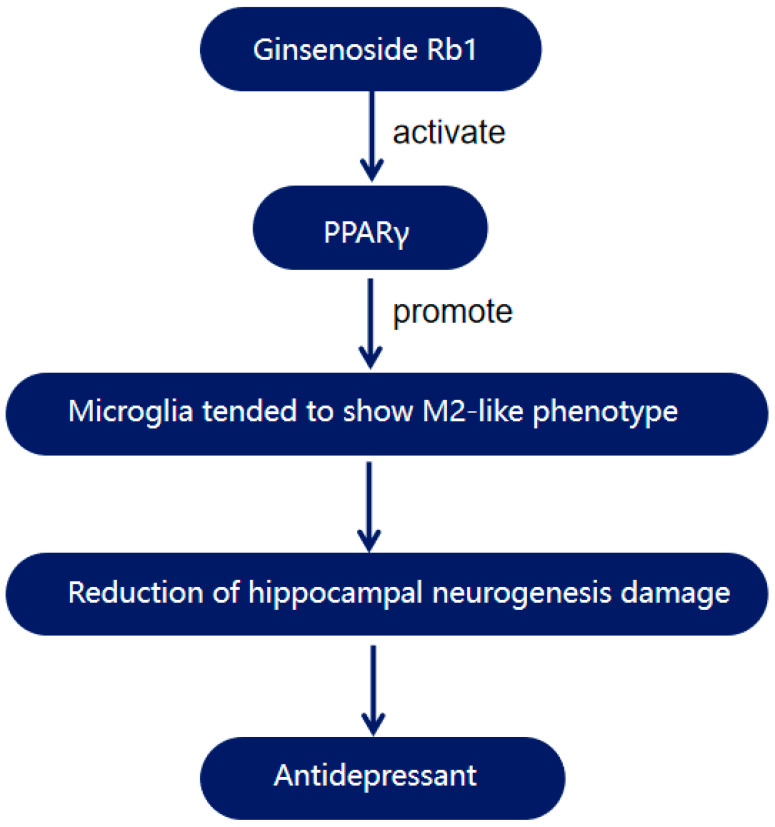
The ginsenoside Rb1 antidepressant pathway through the activation of PPARγ.

**Table 1 antioxidants-14-00238-t001:** Comparison of the effects of ginsenoside Rb1 and the traditional antidepressant fluoxetine on monoamine neurotransmitter levels in the rat brain after treatment in chronic unpredictable mild stress (CUMS) model rats.

Group	Dose (mg/kg)	5-HT	Norepinephrine	Dopamine
Fluoxetine	10	197.3 ± 9.1	134.0 ± 8.8	48.4 ± 5.1
	4	154.3 ± 7.8	98.3 ± 7.5	19.4 ± 9.5
Ginsenoside Rb1	8	171.1 ± 8.7	109.1 ± 11.9	32.8 ± 8.6
	16	191.0 ±8.9	132.0 ±12.0	36.4 ± 4.9

**Table 2 antioxidants-14-00238-t002:** Antidepressant effects and mechanisms of action of ginsenoside Rb1 and conventional antidepressant drugs, as well as natural antidepressant components.

Drug	Regulates Neuroendocrine System	Inhibits Inflammatory Cytokine Production	Inhibits 5-HT Reuptake	Inhibits Noradrenaline Reuptake	Inhibits Monoamine Oxidase	Side Effects	References
Fluoxetine	Yes	Yes	Yes	No	No	Nausea, insomnia, sexual dysfunction, etc.	[46,47,48]
Paroxetine	Yes	Yes	Yes	No	No	Nausea, somnolence, dizziness, etc.	[49,50]
Venlafaxine	Yes	Yes	Yes	Yes	No	Hypertension, tachycardia, nausea, etc.	[51,52]
Amitriptyline	Yes	Yes	Yes	Yes	No	Dry mouth, constipation, blurred vision, etc.	[53,54]
Phenelzine	Yes	Not clear	Not clear	Not clear	Yes	Hypertension, tachycardia, headache, etc.	[55,56]
Ginsenoside Rb1	Yes	Yes	Not clear	Not clear	Not clear	Mild gastrointestinal discomfort, etc.	[37,43,57]
Glycyrrhizin	Yes	Yes	Not clear	Not clear	Not clear	Fewer side effects reported	[58]
Asiaticoside	Yes	Yes	Not clear	Not clear	Not clear	Not clear	[59]
Saikosaponin A	Yes	Yes	Not clear	Not clear	Not clear	Not clear	[60,61]
Kaempferol	Yes	Yes	Not clear	Not clear	Not clear	Not clear	[62]
Cannabidiol	Yes	Yes	Not clear	Not clear	Not clear	Drowsiness, dry mouth, hypotension, etc.	[63]
Icariin	Yes	Yes	Not clear	Not clear	Not clear	Not clear	[64]
Resveratrol	Yes	Yes	Not clear	Not clear	Not clear	Mild gastrointestinal discomfort, etc.	[65]
Ferulic Acid	Yes	Yes	Not clear	Not clear	Not clear	Not clear	[66]
Honokiol	Yes	Yes	Not clear	Not clear	Not clear	Not clear	[67]
Emodin	Yes	Yes	Not clear	Not clear	Not clear	Diarrhoea, abdominal pain, etc.	[68]
Puerarin	Yes	Yes	Not clear	Not clear	Not clear	Not clear	[69]
Curcumin	Yes	Yes	Not clear	Not clear	Not clear	Gastrointestinal discomfort, allergies, etc.	[70]
Ergosterol Peroxide	Yes	Not clear	Not clear	Not clear	Not clear	Not clear	[71]
Berberine Hydrochloride	Yes	Yes	Not clear	Not clear	Not clear	Nausea, vomiting, diarrhoea, etc.	[72]
Ginsenoside Rg1	Yes	Yes	Not clear	Not clear	Not clear	Fewer side effects reported	[73]
Salidroside	Yes	Yes	Not clear	Not clear	Not clear	Not clear	[74]
Piperine	Yes	Yes	Not clear	Not clear	Not clear	Gastrointestinal irritation, etc.	[75]
Proanthocyanidins	Yes	Yes	Not clear	Not clear	Not clear	Not clear	[76]
Rosmarinic Acid	Yes	Yes	Not clear	Not clear	Not clear	Not clear	[77]
β-Asarone	Yes	Not clear	Not clear	Not clear	Not clear	Not clear	[78]
Paeoniflorin	Yes	Yes	Not clear	Not clear	Not clear	Not clear	[79]

## Data Availability

All data supporting the findings of this study are available within the manuscript.

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
