# Peer review of "Pharmacological Mechanism and Drug Research Prospects of Ginsenoside Rb1 as an Antidepressant"

_antioxidants, 2025, doi:10.3390/antiox14020238_

Round 1

Reviewer 1 Report

The manuscript effectively reviews a broad range of mechanisms through which ginsenoside Rb1 may exert its antidepressant effects. The paper includes an in-depth discussion of signaling pathways, neurobiological impacts, and pharmacological profiles. Focus on ginsenoside Rb1 as a natural alternative to traditional antidepressants is timely. Moreover, the integration of pharmacology, neurobiology, and traditional medicine perspectives is impactful.

Overall, the manuscript occasionally feels overly dense with technical details, which could make it hard for readers to follow. The authors are encouraged to use subheadings and bullet points more effectively to break down complex mechanisms and results. Improvements in clarity, focus, and translational discussion would enhance the manuscript's clarity and quality impact. Addressing the following points during revision can significantly strengthen the manuscript's contribution to the field:

General:

While the review is comprehensive, the novelty of the findings about ginsenoside Rb1's antidepressant mechanisms is unclear. Therefore, it is strongly recommended that the authors highlight what new insights this review brings compared to existing reviews.

The low bioavailability of ginsenoside Rb1 is mentioned but not adequately explored, despite it being a critical limitation for clinical applications. Please expand the discussion on strategies to overcome this limitation, such as novel drug delivery systems or combination therapies.

The comparison between ginsenoside Rb1 and traditional antidepressants could be strengthened with more quantitative data or detailed evidence. Please include comparative data on efficacy, side effects, and mechanisms of action, if available. If not, at least this can be communicated which would be a good idea for the next research.

The paper lacks a detailed discussion of how preclinical findings translate into clinical contexts. Please add a section discussing clinical trial prospects, existing studies, and potential hurdles for ginsenoside Rb1 in human applications.

Section-by section:

Abstract: The abstract is overly dense and tries to pack too much information without clearly emphasizing the novelty of the work. While mechanisms of ginsenoside Rb1 are outlined, the clinical relevance and potential advantages over traditional treatments are not highlighted sufficiently. Please streamline the abstract to focus on the key findings, novel insights, and clinical implications. For instance, explicitly state what makes ginsenoside Rb1 a superior or complementary option to existing antidepressants.

Main part: The introduction effectively sets the stage but is heavily descriptive and lacks a clear research gap. For example, while the prevalence and burden of depression are discussed, the rationale for focusing on ginsenoside Rb1 as opposed to other ginsenosides or natural compounds is not well established. Please add a section explicitly stating the research gap and why ginsenoside Rb1 is a compelling candidate. Compare it briefly with other ginsenosides or natural compounds to justify the review's focus.

The discussion on genetic factors is limited and does not sufficiently integrate recent advancements, such as genome-wide association studies (GWAS) or specific gene candidates linked to depression. Please include more recent references to GWAS or epigenetic studies that provide a deeper understanding of genetic influences on depression.

The inflammatory hypothesis is discussed, but there is no clear connection to how ginsenoside Rb1 interacts with inflammation pathways in comparison to other anti-inflammatory interventions. Please provide a specific example or study that demonstrates ginsenoside Rb1's superiority or uniqueness in modulating inflammation compared to standard anti-inflammatory treatments or other natural compounds.

The manuscript emphasizes the effects of ginsenoside Rb1 on serotonin, dopamine, and GABA systems but fails to compare these effects quantitatively with existing antidepressants. Please incorporate comparative data or at least discuss the magnitude of these effects relative to commonly used drugs like SSRIs or SNRIs.

While the antioxidant effects of ginsenoside Rb1 are well-described, there is little discussion of how these effects translate into clinical antidepressant outcomes. Please include preclinical or clinical data showing whether antioxidant effects correlate with behavioral improvements in models of depression.

The bioavailability and delivery challenges section briefly mentions ginsenoside Rb1's low bioavailability but does not explore potential solutions beyond general statements about delivery methods. Please provide specific examples of current research into improving bioavailability, such as nanoparticle delivery systems, prodrugs, or combination therapies with gut microbiota modulators.

The manuscript highlights that ginsenoside Rb1 has fewer side effects but does not quantify or substantiate this claim with concrete data. Please include data from animal studies or clinical trials (if available) that compare the side effect profiles of ginsenoside Rb1 with traditional antidepressants. If not available, suggest these for future research at least.

Conclusion: The conclusion reiterates the potential of ginsenoside Rb1 but does not adequately address its limitations or future directions beyond improving bioavailability. Please consider adding a couple of specific areas for future research, such as identifying which subgroups of patients with depression might benefit most or exploring combinatory use with other treatments.

Author Response

Comments1:Abstract: The abstract is overly dense and tries to pack too much information without clearly emphasizing the novelty of the work. While mechanisms of ginsenoside Rb1 are outlined, the clinical relevance and potential advantages over traditional treatments are not highlighted sufficiently. Please streamline the abstract to focus on the key findings, novel insights, and clinical implications. For instance, explicitly state what makes ginsenoside Rb1 a superior or complementary option to existing antidepressants.

Resprose 1:Regarding the issue you mentioned, I have made modifications to the abstract and highlighted the changes in the review. I believe that the potential advantage of ginsenoside Rb1 lies in its multi-target action against depression. For a disease like depression with a complex pathogenesis, multi-faceted treatment is more advantageous compared to the treatment of traditional antidepressants that target a single site. As for the clinical significance of ginsenoside Rb1, there is currently limited research. I think this may be related to its long action cycle and complex mechanisms. And this is precisely where future research needs to delve deeper.

Lines 13-27

Comments2: The introduction effectively sets the stage but is heavily descriptive and lacks a clear research gap. For example, while the prevalence and burden of depression are discussed, the rationale for focusing on ginsenoside Rb1 as opposed to other ginsenosides or natural compounds is not well established. Please add a section explicitly stating the research gap and why ginsenoside Rb1 is a compelling candidate. Compare it briefly with other ginsenosides or natural compounds to justify the review's focus.

Resprose 2:The selection of ginsenoside Rb1 as a potential antidepressant for this review is based on two main considerations. Firstly, in comparison with components such as resveratrol, there is a relative paucity of systematic summaries regarding ginsenoside Rb1. Ginsenoside Rb1 can elicit antidepressant effects by inhibiting the production of inflammatory mediators and reducing the generation of pro-inflammatory factors. Additionally, it can exert antidepressant actions through other mechanisms. Secondly, Rb1 was chosen over Rg1 or other ginsenosides because the metabolite F2 of ginsenoside Rb1 has been found to possess excellent potential antidepressant effects in recent years. Consequently, I am of the view that this component holds greater research prospects.For a more detailed description, please refer to Table 2 later, where I compare it with other components.

Lines 74-87

Lines 356

Comments3:The discussion on genetic factors is limited and does not sufficiently integrate recent advancements, such as genome-wide association studies (GWAS) or specific gene candidates linked to depression. Please include more recent references to GWAS or epigenetic studies that provide a deeper understanding of genetic influences on depression.

Resprose 3:Regarding the mechanism of action of depression (including the influence of genetic factors), I have combined the opinions of other reviewers and browsed through some literature. I found that others have provided more detailed descriptions of this part. Therefore, I have decided to delete this part and focus on reviewing the mechanism of action of ginsenoside Rb1. I have also consulted GWAS-related literature. Thank you for your comments, which will also be of great help to my future research.

Comments4:The inflammatory hypothesis is discussed, but there is no clear connection to how ginsenoside Rb1 interacts with inflammation pathways in comparison to other anti-inflammatory interventions. Please provide a specific example or study that demonstrates ginsenoside Rb1's superiority or uniqueness in modulating inflammation compared to standard anti-inflammatory treatments or other natural compounds.

Resprose 4:I have added the interaction pathways between ginsenoside Rb1 and inflammatory pathways. For the comparison with other components, I have placed it in Table 2 in the form of a table.

Lines 235-249

Lines 356

Comments5:The manuscript emphasizes the effects of ginsenoside Rb1 on serotonin, dopamine, and GABA systems but fails to compare these effects quantitatively with existing antidepressants. Please incorporate comparative data or at least discuss the magnitude of these effects relative to commonly used drugs like SSRIs or SNRIs.

Resprose 5:I have added this part of the data and marked it in the article.

Lines 263-281

Comments6:While the antioxidant effects of ginsenoside Rb1 are well-described, there is little discussion of how these effects translate into clinical antidepressant outcomes. Please include preclinical or clinical data showing whether antioxidant effects correlate with behavioral improvements in models of depression.

Resprose 6:I was unable to locate any clinical data pertaining to the antidepressant effects of ginsenoside Rb1. Nevertheless, I am of the view that the anti-inflammatory effects exhibited by ginsenosides in the clinical data related to the treatment of chronic kidney disease can, from an anti-inflammatory standpoint, indirectly substantiate the antidepressant effects of ginsenoside Rb1. Consequently, I have incorporated this additional data into the review. Regrettably, the experiment failed to mention the potential side effects of ginsenoside Rb1. Lines 322-339

Comments7:The bioavailability and delivery challenges section briefly mentions ginsenoside Rb1's low bioavailability but does not explore potential solutions beyond general statements about delivery methods. Please provide specific examples of current research into improving bioavailability, such as nanoparticle delivery systems, prodrugs, or combination therapies with gut microbiota modulators.

Resprose 7:I have added content related to improving the bioavailability of ginsenoside Rb1. This includes the delivery of ginsenoside Rb1 by mannose-modified bovine serum albumin drug-loaded nanoparticles, as well as the impact of the interaction between the gut microbiota and ginsenoside Rb1 on its absorption.

Lines 358-400

Comments8:The manuscript highlights that ginsenoside Rb1 has fewer side effects but does not quantify or substantiate this claim with concrete data. Please include data from animal studies or clinical trials (if available) that compare the side effect profiles of ginsenoside Rb1 with traditional antidepressants. If not available, suggest these for future research at least.

Resprose 8:I didn't find any experimentally verified side effects of ginsenoside Rb1. However, ginsenoside Rb1 may cause gastrointestinal discomfort. Therefore, in the conclusion, I put forward suggestions for future research directions. In-depth studies on its administration methods and investigations into the effects of its interaction with the gut microbiota may help discover the potential side effects of ginsenoside Rb1.

Comments9:Conclusion: The conclusion reiterates the potential of ginsenoside Rb1 but does not adequately address its limitations or future directions beyond improving bioavailability. Please consider adding a couple of specific areas for future research, such as identifying which subgroups of patients with depression might benefit most or exploring combinatory use with other treatments.

Resprose 9:

Lines 428-456

Reviewer 2 Report

The authors present a detailed review of the potential for the use of ginsenoside Rb1 in the pharmacotherapy of major depressive disorder. They underpin this with details of the pharmacological mechanism of this naturally occurring compound. The review is well illustrated. I have no major comments.

Lines 50-1: Promethazine is not generally considered to be a tricyclic antidepressant, even though it does indeed have a tricyclic molecular structure.

Line 357: Since the SI unit of energy is the joule, it would be good if the figure given as kcal/mol could also be stated in J/mol.

Author Response

Comments1:Lines 50-1: Promethazine is not generally considered to be a tricyclic antidepressant, even though it does indeed have a tricyclic molecular structure.

Resprose 1:

Lines 49-52

Comments2:Line 357: Since the SI unit of energy is the joule, it would be good if the figure given as kcal/mol could also be stated in J/mol.

Resprose 2:Thank you for your corrections.

Lines 220

Reviewer 3 Report

This manuscript provides a comprehensive review of the antidepressant potential of ginsenoside Rb1, focusing on its mechanisms of action, including antioxidant, anti-inflammatory, and neuroprotective properties. The authors successfully highlight the therapeutic potential of this compound, offering insights into its pharmacological effects and possible applications. However, there are several areas where the manuscript must be improved to enhance clarity, coherence, and academic rigour.

The abstract and conclusion are overly optimistic and do not adequately reflect the challenges like bioavailability and the need for clinical validation

The section on the pathogenesis of depression (section 2) is unnecessary for the manuscript's purpose and lacks novelty. Removing this section (with both figures) would streamline the focus on ginsenoside rb1.

The review will significantly benefit if any clinical evidence or information about the lack of such will be introduced into the discussion section of the manuscript.

Figure 4 (PPARγ) is underutilized in the narrative and poorly integrated. Reference these figures explicitly in the text where the mechanisms are discussed to improve coherence.

I suggest an alternative arrangement for Table 1:

G Rb1

Venlafaxine

Resveratrol

….

Inhibits reuptake of 5-HT

X

X

Anti-inflammatory

X

Regulates neuroendocrine system

X

 I would also suggest significantly expanding the list of antidepressant drugs in this table—this would make it an interesting source of knowledge.

The comparison with other natural antidepressants is informative but lacks depth. Expand on the pharmacological distinctions between ginsenoside Rb1 and other natural compounds, especially regarding their mechanisms and clinical applicability.

Expand the limitations section and discuss future research directions.

Some references are outdated (ref 9, 11 ...). Some are are inconsistently formatted (e.g. 10). Focus on newer sources.

none

Author Response

We thank you for your thoughtful suggestions and insights.The manuscript has been rechecked and the necessary changes have been made in accordance with your suggestions. The responses to all comments have been prepared and are attached herewith.

Comments1:The abstract and conclusion are overly optimistic and do not adequately reflect the challenges like bioavailability and the need for clinical validation

Resprose 1:I have highlighted the modified parts in red and marked the line numbers. Regarding the abstract and conclusion you mentioned, I have made modifications and added possible directions for improving bioavailability in the main content.

Lines 13-27

Lines 428-457

Lines 358-400

Comments2:The section on the pathogenesis of depression (section 2) is unnecessary for the manuscript's purpose and lacks novelty. Removing this section (with both figures) would streamline the focus on ginsenoside rb1.

Resprose 2:Thank you for your suggestions. I have deleted the second part and adjusted the structure of the article.

Comments3:The review will significantly benefit if any clinical evidence or information about the lack of such will be introduced into the discussion section of the manuscript.

Resprose 3:I was unable to locate any clinical data pertaining to the antidepressant effects of ginsenoside Rb1. Nevertheless, I am of the view that the anti-inflammatory effects exhibited by ginsenosides in the clinical data related to the treatment of chronic kidney disease can, from an anti-inflammatory standpoint, indirectly substantiate the antidepressant effects of ginsenoside Rb1. Consequently, I have incorporated this additional data into the review. Regrettably, the experiment failed to mention the potential side effects of ginsenoside Rb1. Lines 322-339

Comments4:Figure 4 (PPARγ) is underutilized in the narrative and poorly integrated. Reference these figures explicitly in the text where the mechanisms are discussed to improve coherence.

Resprose 4:I have reorganized the content related to Figure 4.

Lines 235-249

Comments5:I suggest an alternative arrangement for Table 1:

G Rb1

Venlafaxine

Resveratrol

….

Inhibits reuptake of 5-HT

X

X

Anti-inflammatory

X

Regulates neuroendocrine system

X

 I would also suggest significantly expanding the list of antidepressant drugs in this table—this would make it an interesting source of knowledge.The comparison with other natural antidepressants is informative but lacks depth. Expand on the pharmacological distinctions between ginsenoside Rb1 and other natural compounds, especially regarding their mechanisms and clinical applicability.

Resprose 5:I have made modifications to the format and supplemented the content of the table, as detailed below.

Lines 356

Comments6:Expand the limitations section and discuss future research directions.

Resprose 6:I believe that future research directions should focus on combination therapies for depressive symptoms caused by other diseases. For example, the depressive behaviors induced by diabetes. On the other hand, combination treatments with other natural ingredients whose mechanisms of action complement those of this component can be explored for treating depression. Additionally, more research on drug safety is needed.

I have supplemented this part of the content in the conclusion.

Comments7:Some references are outdated (ref 9, 11 ...). Some are are inconsistently formatted (e.g. 10). Focus on newer sources.

Resprose 7:I have updated some of the references.

Round 2

Reviewer 1 Report

The authors have carefully taken into consideration the points and have accordingly revised the manuscript. Thanks. There are no further comments.

There are no further comments.

Author Response

Thank you for your thoughtful advice and insights.

Reviewer 2 Report

I thank the authors for kindly responding to my previous comments.

Not applicable.

Reviewer 3 Report

Overall, I am satisfied with the corrections made however, I still have some minor comments:

Line 236 – When the Authors introduce CMS, please specify that it is an animal model.

Table 1 – Missing space in "98.3±7.5".

Line 323 – "Previous studies report limited clinical experimental..." – Please provide references for these "previous studies."

Table 2 – Reduce the font size and decrease the spacing between rows.

Line 381 – Provide full names for the abbreviations: Rd, F2, Rg3, CK, and PPD.

Lines 384-388 – Provide references for this section.

Lines 390-392 – Provide references for this section.

Provide the full name for ROS and ensure that every abbreviation is spelt out in full upon first use.

Lines 433-435 – "There are few reports on novel..." – Please provide references for these reports.

Round 3

Reviewer 3 Report

I am satisfied with correction made.

none